# The changing meaning of "no" in Canadian sex work

**Lynn Kennedy** [ORCID] *

Sex Work Population Project https://populationproject.ca/, Vancouver, British Columbia, Canada

* lynn.kennedy@populationproject.ca

**Data Availability Statement:** All code and data may be found here https://osf.io/hwzsn/ Code and data are publicly available.

**Funding:** The author(s) received no specific funding for this work.

## Abstract

With the migration of sex workers to online advertising in Canada, a substantial body of research has emerged on how they communicate with prospective clients. However, given the enormous quantity of archival material available, finding representative ways to identify what sex workers say is a difficult task. Numerical analysis of commonly used phrases allows for the analysis of large numbers of documents potentially identifying themes that may be missed using other techniques. This study considers how Canadian sex workers communicate by examining how the word "no" was used by online advertisers over a 15-year period. Source materials consisted of three collections of online classified advertising containing over 4.2 million ads collected between 2007 and 2022 representing 214456 advertisers. Advertisers and demographic variables were extracted from ad metadata. Common terms surrounding the word "no" were used to identify themes. The word "no" was used by 115127 advertisers. Five major themes were identified: *client reassurance* (54084 advertisers), *communication* (47130 advertisers), *client race* (32612 advertisers), *client behavior* (23863 advertisers), and *service restrictions* (8545 advertisers). The probability of there being an association between an advertiser and a major theme was found to vary in response to several variables, including: time period, region, advertiser gender, and advertiser ethnicity. Results are compared with previous work on race and risk messaging in sex work advertising and factors influencing *client race* restrictions are considered. Over time, the restriction related themes of *client behavior*, *service restrictions*, and *client race* became more prominent. Collectives, multi-regional, cis-female, and Black or Mixed ancestry advertisers were more likely to use restrictions.

## Introduction

Researchers describe sex workers as a hidden population [1]. However, online advertising has provided an opportunity to gain insights into this population and a substantial body of research now exists examining sex workers' online communication practices [2–13]. Given the enormous quantity of archival data available, finding representative ways to understand how sex workers communicate is a difficult task. In this study, I consider how three large cohorts of Canadian sex work advertisers communicated with prospective clients over a span of 15 years.

**Competing interests:** The authors have declared that no competing interests exist.

In particular, I consider how the commonly used word "no" was employed in the context of online advertising.

## Research at scale

For the purposes of this analysis, "at scale" refers to studies that represent over 1000 workers. Document analysis at scale for the purposes of law enforcement is common. See Dimas et al. [14] for a review. Outside the law enforcement context, research that studies the sociology of sex work using large document collections appears to have a wide focus: Cunningham and Kendall [15] analyzed metadata from over 94000 sex worker profiles on one popular US review site to better understand health and safety practices; Nelson et al. [16] in a study of factors affecting hourly rates looked at an initial sample of 1730 independent escorts; Griffith et al. [10] reviewed 2925 advertisements of female sex workers comparing advertised physical attributes with fees charged; Kingston and Smith [11] enumerated gender and sexual orientation of sex workers in the UK, reviewing 25511 escort directory member profiles; Chan et al. [17] used 16735 online profiles to identify changes in county-level prostitution activity as a popular US online classifieds site expanded; lastly, Kennedy [18] used a database of over 3.6 million Canadian online classified ads to examine how sex worker populations change over time.

The current study undertakes a numerical analysis of commonly used phrases in ad texts. The evidence presented here partly supports the findings of Moorman and Harrison [8] who, in a study of Backpage.com, provided evidence that Black advertisers used risk mitigating language more often than White advertisers.

The motivation for this study emerged from the process of characterizing systematic error in Kennedy 2022 [18]. This required reading thousands of ads, where phrases starting with "no . . ." would appear quite frequently. Many of these statements appeared to refer to restrictions, short imperative statements such as "no explicit talk", that set ground rules for the interaction between the advertiser and prospective clients. These restrictions can relate to health related messaging [7] but also, as will be shown in this study, encompass other criteria relating to physical, material and psychological safety.

The classified advertising sites available as source material typically segment ads based on service, gender, and location (city, provincial region, or territorial region). Advertisers place ads in specific segments where the ad will be visible on the main listing page for that segment. The ads tend to be temporary. Advertisers have to either repost ads or pay a fee to "bump" the ad for it to be visible on the main page. How long an ad is visible can depend on whether the advertiser deletes the ad, competing advertisers posting ads, and content moderation (either by the community or the site operator). For understanding advertising behavior, this time locality is crucial as it provides a clear indication when an advertiser was active.

In addition to ad text, ad metadata, or common data extracted from ads, can be identified. In general, this metadata can identify advertisers, authors of groups of ads that may represent one or more workers, and can tell us where the advertiser is advertising, what gender and, sometimes, ethnicity they self-identify as. This study combines this metadata with language use to better understand associations between how advertisers used "no" and these demographic variables.

## Historical context and safety

In this study, safety is defined as "a state in which hazards and conditions leading to physical, psychological or material harm are controlled in order to preserve the health and well-being of individuals and the community" [19]. This definition is used because, as will be seen in the

analysis below, the concept of safety in advertising extends beyond freedom from physical harm to include the mitigation of other types of risk.

In contrast to street based workers, where negotiations tend to be brief [20, 21], online advertisers typically have more time to interact with prospective clients. As a safety strategy, advertising online significantly reduces risk for sex workers [3, 20, 22]. Statements that set boundaries are an important part of how these workers communicate with prospective clients. In this study a "restriction" refers to any statement that specifically limits services provided, who can contact the advertiser, or how they can interact with them.

Workers describe investing considerable effort in getting to know clients before any in-person contact is initiated [3, 5, 23, 24] and this can include more formal screening processes in some cases [3, 13, 22, 23, 25]. Information sharing between workers is another important safety strategy [13, 26] and how workers share information has changed as the internet has become the dominant way that workers advertise [3, 23, 26]. Overall workers who use online advertising engage in less risky behavior [15] and experience decreased stress and reduced workplace violence [20]. These workers can enjoy more privacy than, for example, street based workers but may also be subject to unique risks relating to malicious online behavior by clients or competitors [5].

This study focuses on sex workers who provide sexual services in person termed *contact* sex workers. In Canada, these workers have been subject to increasing criminalization [27, 28]. Starting in 1892 female sex workers in Canada were targeted by common bawdy house and vagrancy sections of the criminal code [27, 28]. Nevertheless, by the late 1960s, indoor sex work was generally tolerated [27]. However, in the late 1970s moral panic over the murder of Emmanuel Jaques in Toronto resulted in the closure of most licensed indoor venues in Canada [27]. At around the same time the Hutt decision hampered police efforts to arrest sex workers who worked outdoors. This resulted in many workers migrating to street prostitution and escort agencies [27, 28]. The federal government responded by criminalizing public communication for the purposes of prostitution in 1985 [28]. These events precipitated significant increases in the incidence of violence against sex workers in the late 1980s and 1990s [29, 30].

Technology was to change this situation. In 2001 Craigslist.com, a free classifieds site, launched in Canada [31]. Craigslist's "erotic services" section provided a free means for sex workers to market services online. The adoption of the site was rapid (see supporting information S1 Appendix). Sex worker populations are dynamic [18, 32] and there is evidence that new populations of workers entered the industry with the advent of online classified advertising [33].

Under pressure from the Canadian government, Craigslist.com was forced to remove the erotic services section in 2010 [34]. In its wake Backpage.com became the dominant venue for sex work advertising in Canada [35]. In 2014 the Protection of Communities and Exploited Persons Act (PCEPA) [36] was enacted in Canada effectively outlawing online advertising and criminalizing the purchase of sexual services. Four years later in 2018 Backpage.com succumbed to legal pressure culminating in its seizure by the FBI [37]. In Canada, following the demise of Backpage.com, online classified advertising largely consolidated on one remaining site still used by sex workers today [38 preprint].

The closure of websites not only directly affected workers around the world [39–41] but also made it more difficult for advocacy organizations [13] and workers to establish online communities and mentoring [42]. However, in Canada, criminalization appears to have had little practical effect on most workers with the exception of im/migrant workers who experienced worse outcomes [43].

From a theoretical perspective online advertising provides one part of an 'enabling environment' that sex workers employ to manage risk [44, 45]. As will be shown below, and as

Moorman and Harrison [8] describe, intersectionality [46] is important for understanding how advertisers use this environment.

### Research questions and objectives

This study aimed to explore 1) the evolution of the use of the word "no" and other terms relating to safety and racial discrimination over time, and 2) whether the use of "'no" differs by the demographic characteristics of the advertisers in online sex work advertisements in Canada from 2007–2022.

## Materials and methods

This research considers how common themes in sex work advertising relate to demographic variables surrounding how the word "no" is used. The probability of there being an association between an advertiser using "no" and a major theme is calculated for several variables, including: time period, region, advertiser gender, and advertiser ethnicity.

### Extracting ad data

Source materials for the analysis were advertising texts from online classified ads. Ads were collected from six prominent advertising sites during three time periods: April 1, 2007 to March 31, 2009 inclusive, November 1, 2014 to December 31, 2016 inclusive, and September 15, 2021 to September 22, 2022 inclusive. The sites in question were well known advertising venues for sex work in Canada as determined by a group of experts from the *Sex*, *Power*, *Agency*, *Consent*, *Environment and Safety* Project (SPACES) [47]. SPACES was initiated in 2012 at the University of British Columbia to explore health and safety issues experienced by off-street sex workers.

All sites used similar navigation for classified ads, described in the Introduction. To find ads, pages listing ads by region, gender, and service were downloaded and scanned for ad URLs. In addition, *sitemaps* or lists of ad URLs provided by site operators were used to identify recent ads. Ads were downloaded multiple times per hour for the 2007–2009 and 2014–2016 collections, and ads from sitemaps were downloaded multiple times per day for the 2021–2022 collection.

The ad collections for 2014–2016 and 2021–2022 included ads from all provinces and territories in Canada. The 2021–2022 data was limited to one year as the source site stopped updating the sitemaps that were being used to retrieve ads in September, 2022. The 2007–2009 collection included ads from British Columbia (BC) only. This collection was included because there is little available historical data from this period and how the advertisers in this region communicated changed substantially.

This study uses the abstract entity of an *advertiser* (Kennedy, 2022) as a meaningful way to group ads. An advertiser represents the author of a distinct group of ads in each collection. Advertisers, who could represent one or more workers, were identified either from contact information in 2007–2009 and 2014–2016 or internal ids for a chat function in 2021–2022 using methods described in (Kennedy, 2022).

### Defining "no"

For the thematic analysis, ad texts were cleaned by removing most non-alphanumeric characters and were scanned for relevant groups of terms. Pairs of words (bigrams), and triplets of words (trigrams) were extracted along with associated advertiser variables. Variables came from two sources: the ads themselves and ad URLs. In both cases, common fields were used to

identify metadata. All data was stored in a MariaDB database [48] for further analysis. Supporting information S1 File contains a copy of the anonymized database.

In this analysis, frequently used bigrams beginning with "no" were combined with the most common words preceding and following them to generate a relatively small number of meta-documents that could be analyzed for common themes. These files can be found in supporting information S2 File. QualCoder [49] was used to code the files. Theme ranks were calculated by counting the number of advertisers associated with bigrams related to that theme. The thematic analysis was based on three groups of 100 files, one for each collection.

Fig 1 is an example of one of the input files for the bigram "no low". This bigram was included in the *client behavior* code described below, as the embedded usage of the bigram refers to restrictions on negotiating prices. Note that in this case, the words following the bigram are the most salient terms. This was usually the case for most bigrams. In general, the meanings of the bigrams were consistent based on context words.

Thematic analysis as described by Braun and Clarke [50] "involves searching across a data set to find repeated patterns of meaning." In this study, themes were based on the semantic meanings of frequently used phrases in advertising. Advertisers likely assume that the reader knows ad-specific abbreviations and terms. Clients typically learn these terms from other online resources such as review sites [51] and these sites proved to be an invaluable resource for understanding how advertisers communicated. This thematic analysis provides a detailed account of these commonly used phrases and uses intersectionality to interpret the themes. I take a realist approach to understanding the texts: similar texts are assumed to have similar meanings based on how those terms are commonly used in the cultural context of Canadian sex work.

## Statistical measures

The proportion of advertisers associated with each coded theme was calculated and segmented by the following variables: time period, region, self-identified gender, and self-identified ethnicity. These proportions represent the probability that an advertiser fitting a specific demographic category was associated with a given theme (e.g., p(*communication* = True | gender = *male*)). For each theme, the differences between advertiser proportions for each of the demographic variables and the proportions for all advertisers were tested for significance using the

a) 2007-2009                b) 2014-2016                c) 2021-2022

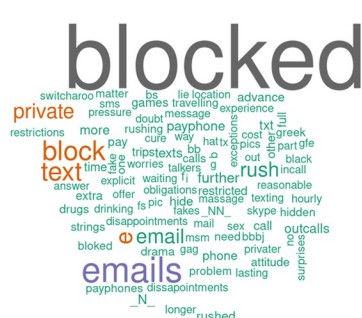

**Fig 1. Example coding file for the bigram "no low" for the 2021–2022 collection.** "Before" and "after" words are terms that preceded and followed the bigram ordered in descending order by advertiser frequency. "_NNNN_" replaces a four-digit number in the anonymized data.

R *prop.test* function (R Core Team, 2021). Pearson correlations between advertisers' use of each of the top five themes were calculated by assigning a 1 if the advertiser used the theme and 0 if not. The R *cor* function was then used to test for associations between each theme.

A key variable, time period, was considered. This is partly because the retrieved materials cover a fifteen year period and it is possible that how advertisers used "no" may have changed during this time. Given that the sites where sex workers could advertise changed over the years as political pressure on site operators grew, the time period roughly coincides with the site. Two analyses were undertaken. For the earliest time period, where only regionally limited data from British Columbia was available, the analysis was limited to this region. A second analysis of data from all regions for 2014–2016 and 2021–2022 was undertaken to identify trends nationally.

For the other variables, with the exception of advertiser ethnicity where data was only consistently available from the 2021–2022 collection, data from all collections was used to calculate probabilities of association. Part of the motivation for this choice is that the technical and economic requirements for advertisers in 2021–2022 were much higher than in the past and I felt that leaving out older advertisers could reduce the generality of the results. Breakdowns of the contributions of each site, including a comparison by gender and site for each of the main five themes, are provided in supporting information S3 File. Results are also compared with an earlier study [8] on race and risk messaging on Backpage.com.

Advertisers, as discussed previously, typically only advertise for relatively short periods of time [18]. This can result in an advertiser sometimes taking on more than one identity, violating the independence of samples. Advertisers associated with multiple identifiers did not appear to be common [18] and the reported results do not attempt to compensate for this violation of independence. Compensation for the use of multiple identifiers did not decrease the probability of an advertiser category being associated with any theme, and did not substantially change the relative associations between advertisers and themes. See supporting information S2 Appendix for a comparison.

Advertiser frequencies for other terms were collected to better understand some emergent themes. To better understand how advertiser communication around safety has changed, advertiser frequencies were calculated for the most common health related bigrams starting with the word "safe" ("safe play", "safe service", "safe services", and "safe gfe"). More recently, advertisers have started screening clients and asking for deposits before meeting them. To provide historic context for this phenomenon, "screening" and "deposit required" were searched as well. To better understand the theme of *client race* described below, the terms "all races", "all nationalities", "all backgrounds", and "all ethnicities" were also searched.

## Ethics statement

All source data used in this study consisted of publicly available data at the time it was collected and was collected in accordance with the policies of the sites in effect at the time. The methods used are conformant with the ethical standards of the Canadian Sociology Association (section 4.10 II) and the American Sociology Association (section 10.5 c) [52, 53]. As the replicability of the main results of this paper is important, a data set is provided as part of the supporting information. However, to protect the safety and privacy of advertisers and third parties, all identifying information has been removed.

## Results

### Collected data

Table 1 summarizes the advertising data in the three collections. A total of 4225069 ads were used (2007–2009: 385729 ads; 2014–2016: 2951642 ads; 2021–2022: 887698 ads). Of these, 39% (N = 1628698) were ads that contained the word "no". Fig 2 shows three word clouds illustrating the relative term frequencies of the words following "no" for each period. More than half of all advertisers used "no" in at least one ad (54%, N = 115127) and these advertisers used mean 2.3 distinct "no" bigrams (SD 2.1) an average of 76% of the time (SD 32%).

### Themes and codes

Table 2 shows the most common associated bigrams for each theme and the number of associated advertisers. A total of 168 unique "no" bigrams were represented in the top 100 bigrams from each period. A total of 14 theme codes were created from the analysis of these bigrams and context words. The vast majority of advertisers using "no" statements are associated with both the top 100 bigrams for each period (92%, N = 105838) and the top 5 codes (88%, N = 100688).

The top 5 themes were coded as *communication*, *client reassurance*, *client behavior*, *client race*, and *service restrictions*. Three of these themes unequivocally related to restrictions: *client behavior*, *client race* and *service restrictions*. Over 51% of advertisers using "no" statements used at least one restriction (51567/100696). On average these advertisers used restrictions in 68% of their ads (SD 36%).

Less common themes used by 2% or fewer advertisers were coded as *services offered*, *no pictures*, *pimps or law enforcement*, *service time*, *employment*, *client age*, *appearance*, and *payment*.

The *communication* theme related to how advertisers wished to be contacted. *Client reassurance* refers to statements such as "no rush" that were intended to reassure prospective clients. *Client behavior* refers to restrictions relating to etiquette such as not negotiating prices, wasting the advertiser's time, or using explicit language. *Client race* refers to advertisers restricting prospective clients based on racial background. In the vast majority of cases, this is a restriction on Black or African American clients. The term *race* is used here because of how clients were typically described in the ads. *Service restrictions* refer to statements regarding disallowed types of services.

Other service related themes were *payment*, *service location*, and *service time* which described restrictions on methods of payment, venues, and service duration respectively. In addition to *client race*, *client age*, and *pimps or law enforcement* are restrictions on who can contact the advertiser and could be said to overlap with *client behavior*. The *no pictures* and *appearance* themes relate to restrictions on presentation in ads and presentation in person, respectively. Finally, *employment* is unique to advertisers looking for prospective employees rather than clients.

**Table 1. Ads and number of advertisers by time period.**

| period | region | sites | ads | | advertisers | |
|---|---|---|---|---|---|---|
| | | | total | using "no" | total | using "no" |
| 2007–2009 | British Columbia | 1 | 385729 | 160458 (42%) | 7939 | 4146 (52%) |
| 2014–2016 | Canada | 6 | 2951642 | 1051167 (36%) | 167539 | 91106 (54%) |
| 2021–2022 | Canada | 1 | 887698 | 417073 (47%) | 38978 | 19875 (51%) |
| All | | 6 | 4225069 | 1628698 (39%) | 214456 | 115127 (54%) |

**Fig 2. Top 100 terms following the word "no" by time period based on term frequency.** Larger words represent more frequently used terms.

The Pearson correlations between the top 5 codes as they were used by advertisers were not high with most not exceeding +/-0.06. However, *client reassurance* was slightly negatively correlated with *client race* (-0.13) and *communication* (-0.16). The *service restrictions* theme was slightly positively correlated with *client behavior* (0.15).

**Table 2. Coded themes relating to the usage of the word "no" based on the top 100 bigrams for each period.** Ambiguous bigrams are followed by the most common succeeding terms in parentheses.

| theme codes | advertisers | common bigrams |
| --- | --- | --- |
| *client reassurance* | 54084 (25%) | "no games", "no disappointments" |
| *communication* | 47130 (22%) | "no blocked" (calls), "no emails", "no text" |
| *client race* | 32612 (15%) | "no black" (gents), "no african", "no aa" |
| *client behavior* | 23863 (11%) | "no lowballers", "no negotiations" |
| *service restrictions* | 8545 (4%) | "no bareback", "no anal" |
| *service location* | 3532 (2%) | "no carcalls", "no outcall" |
| *services offered* | 2786 (1%) | "no limits", "no restrictions" |
| *no pictures* | 2713 (1%) | "no picture", "no face", "no free" (pictures) |
| *pimps or law enforcement* | 2679 (1%) | "no law" (enforcement), "no thugs", "no pimps" |
| *service time* | 1426 (1%) | "no hh", "no half" (hour) |
| *employment* | 988 (<1%) | "no experience" |
| *client age* | 917 (<1%) | "no young" (men) |
| *appearance* | 294 (<1%) | "no tattoos" |
| *payment* | 169 (<1%) | "no e" (transfer) |

## Variables

The following sections describe how the variables of time period, region, advertiser gender, and advertiser ethnicity are associated with the 5 most prevalent codes. See also supporting information S4 File for the numbers of advertisers stratified by variable category and S1 Fig for a graphical comparison of the relative proportions of advertisers associated with each variable category.

**Time period.** Data in 2007–2009 was only available for British Columbia therefore a comparison by time period was made for advertisers who had advertised in BC. Over time, the relative ranks of the top 5 codes changed for BC advertisers. Fig 3 shows the relative ranks of the top 5 codes by time period. With the exception of *communication*, themes became more prominent between 2007–2009 and 2021–2022. Three restriction-related themes had the largest increases during this period: *service restrictions* (13% increase), *client race* (16% increase), and *client behavior* (19% increase).

How do advertisers from all regions in Canada compare? Fig 4 shows the relative ranks of the top 5 codes for periods where data was available from all regions. Comparing Figs 3 and 4, the relative rank of *client race* in 2014–2016 was higher for Canada (third) than it was for BC (fourth). However, *client race* became less prominent in 2021–2022 for Canada (p(*client race*) = 0.12, N = 5397) compared to BC (p(*client race*|BC) = 0.16, N = 1239).

Safety-related messaging generally became more common between 2007 and 2022. The *service restrictions* theme increased both for British Columbia, a 13% increase between 2007 and 2022, and for Canada as a whole, a 12% increase between 2014 and 2022. In addition to the *service restrictions* theme, the usage of the word "safe" itself provided further evidence for this trend. Over all periods 6% (N = 16127) of advertisers used at least one of the tested "safe" bigrams ("safe service", "safe services", "safe gfe" and "safe play"). In 2007–2009 these bigrams were not common with only 1.3% (N = 105) of advertisers using them. However, in 2014–

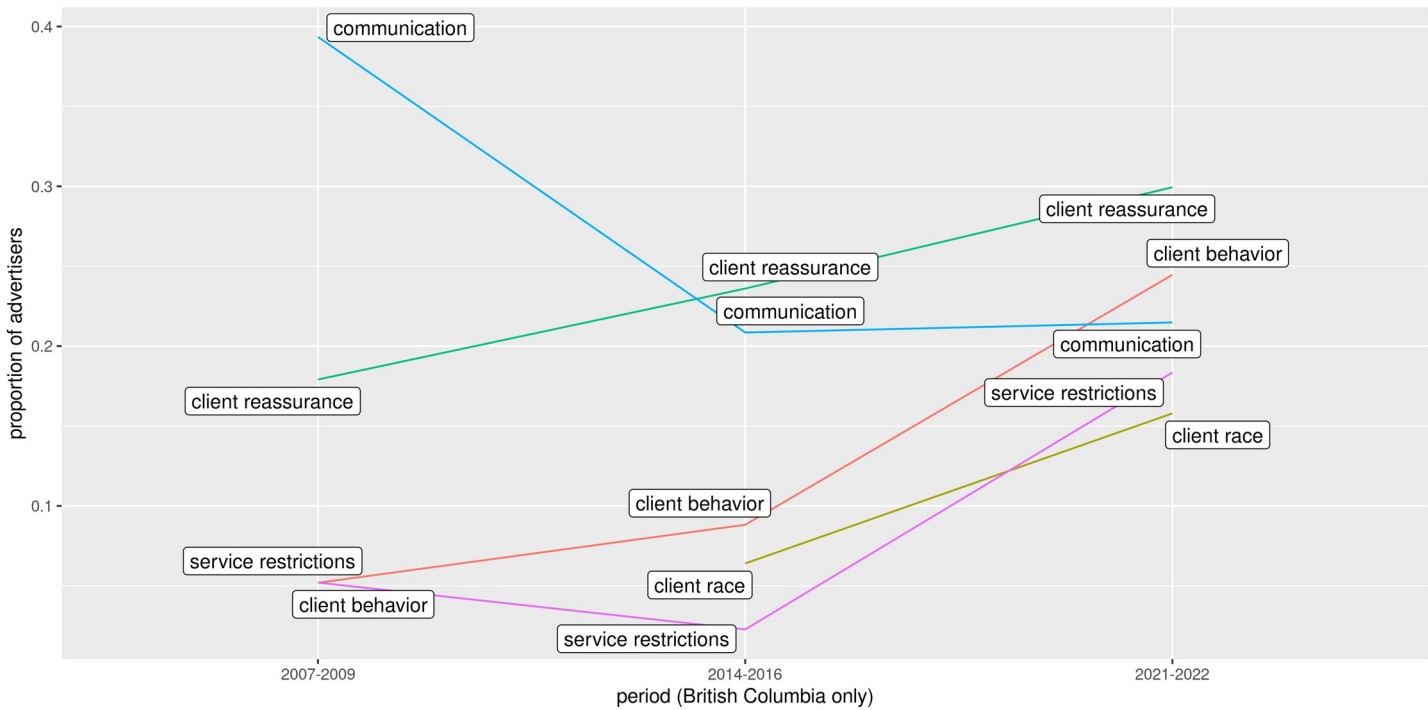

**Fig 3. Advertisers using the top 5 codes by time period in British Columbia.**

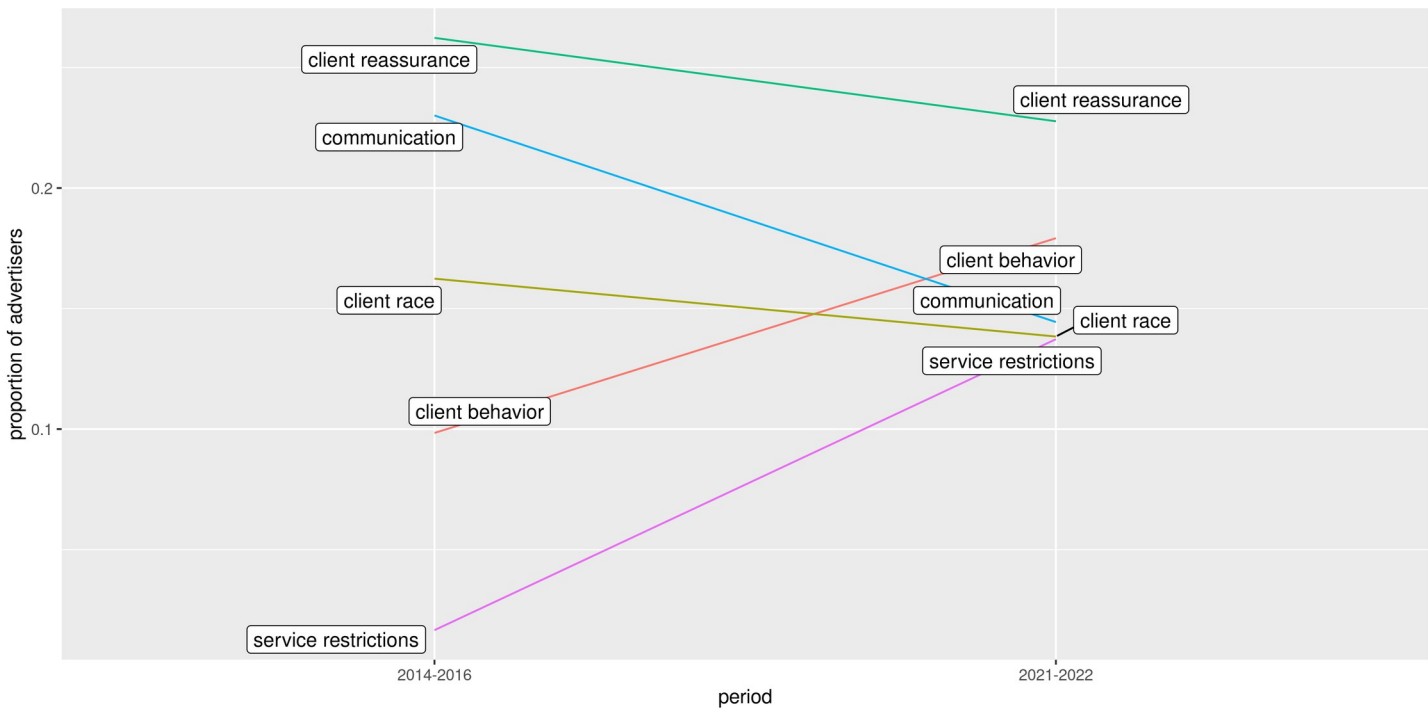

**Fig 4. Advertisers using the top 5 codes by time period for periods including all regions.**

2016 this increased to 7% of advertisers (N = 12166) and 10% (N = 3856) in 2021–2022. Most advertisers using these "safe" bigrams also used "no" (84%, N = 13522 all periods).

Safety messaging was not limited to health. The terms "deposit required" and "screening" were less commonly used but represent important safety strategies. These became more prevalent between 2007 and 2022. In 2007–2009 only 2 advertisers (<1%) used the bigram "deposit required" however by 2021–2022 usage increased to 1725 advertisers (4%). Similarly, the term "screening" in 2007–2009 was used by 8 advertisers (<1%) but in 2021–2022 this grew to 522 (1.3%).

**Region.** Regional differences in code usage are illustrated in Fig 5. Those advertising in *multiple* regions were significantly more likely to be associated with restrictions compared to other advertisers (p(*restriction*|*multiple*) = 0.41, N = 9707 versus p(*restriction*) = 0.24, N = 51567, 95% CI [0.25, 0.26], p < 0.001). *Ontario* advertisers were also more likely to be associated with these codes (p(*restriction*|*Ontario*) = 0.32, N = 25092, 95% CI [0.16, 0.17], p < 0.001).

**Advertiser gender.** Fig 6 shows the relative proportions of advertisers by self-identified gender associated with the top 5 codes. Advertisers self-identifying as cis-female were significantly more likely than other advertisers to be associated with one of the restriction related codes (p(*restriction*|cis-female) = 0.28, N = 48601 versus p(*restriction*|non-cis-female = 0.08, N = 2989, *prop.test* CI [0.15, 0.16], p < 0.001). Cis-male advertisers were the least likely to be associated with restrictions (p(*restriction*|cis-male) = 0.04, N = 671).

**Advertiser ethnicity.** The source site for the 2021–2022 dataset, had an optional "ethnicity" field. Given the context, this field is more akin to the concept of "race" [54] as physical characteristics are the likely differentiator for most categories except Asian Canadian. Fig 7 shows the proportions of advertisers by self-identified ethnicity associated with each of the top 5 codes in 2021–2022. Advertisers associated with multiple ethnicities were significantly more

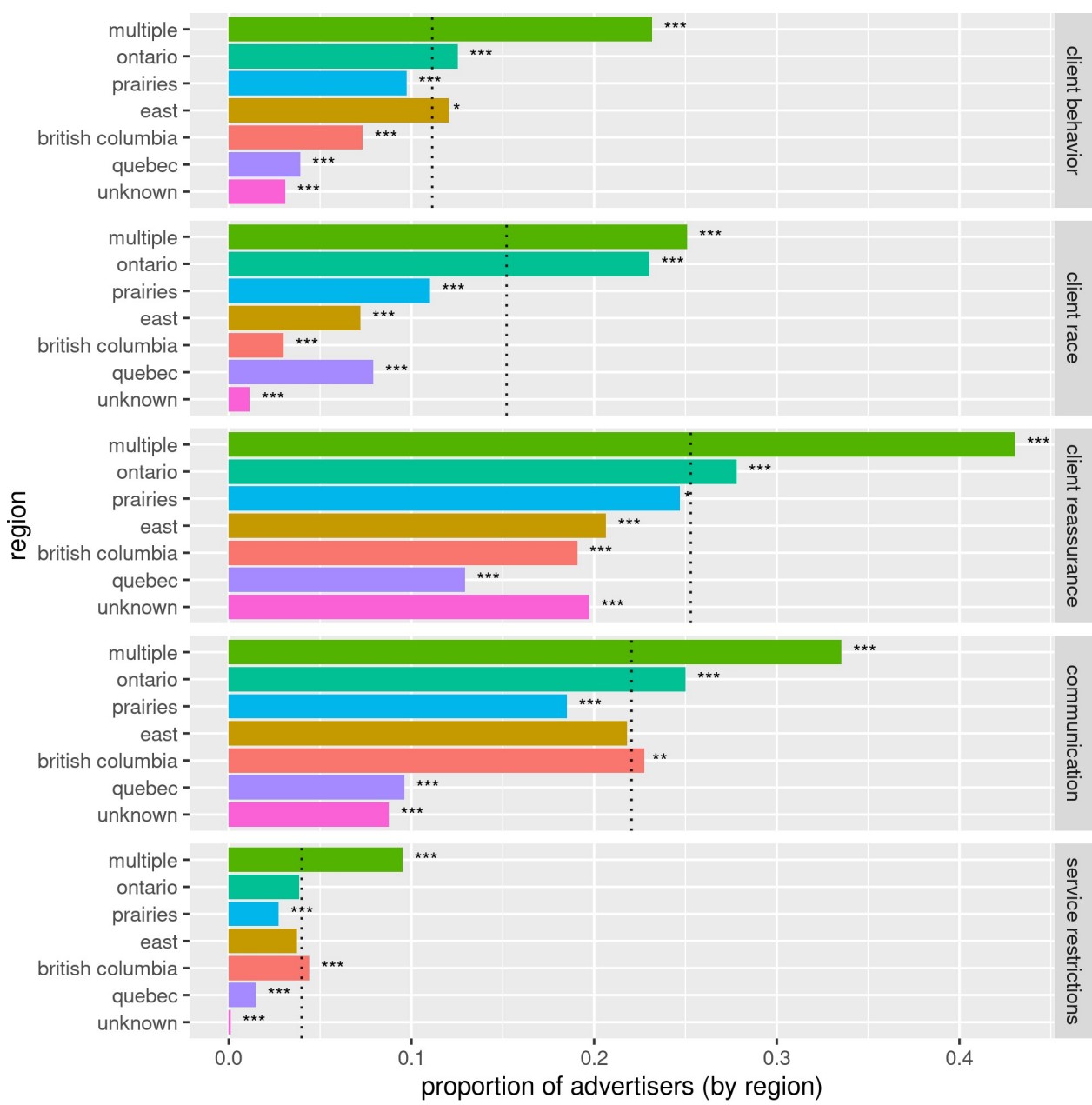

**Fig 5. Advertisers by region associated with the top 5 codes by region.** The dotted lines indicate the proportion of all advertisers associated with each theme. Most regions were significantly different from the overall proportion for each theme: *** p < 0.001, ** p < 0.01, * p < 0.05.

likely to be associated with one of the restriction related themes compared to all advertisers (p (*restriction*|multiple) = 0.93, N = 5905 versus p(*restriction*|not-multiple) = 0.36, N = 11831, 95% CI [0.55, 0.57], p < 0.001). Black and Mixed ancestry advertisers were the only other ethnicities that had significantly higher than overall associations with the three restriction related themes (p(*restriction*) = 0.30; p(*restriction*|Black) = 0.37, 95% CI [0.05, 0.09], p < 0.001; p (*restriction*|Mixed = 0.36, 95% CI [0.04, 0.08], p < 0.001). However, Asian, Asian Canadian, and Indo Canadian all had significantly lower associations than overall associations with the restriction related themes (p(*restriction*|Asian) = 0.24, 95% CI [-0.08–0.05], p < 0.001; p

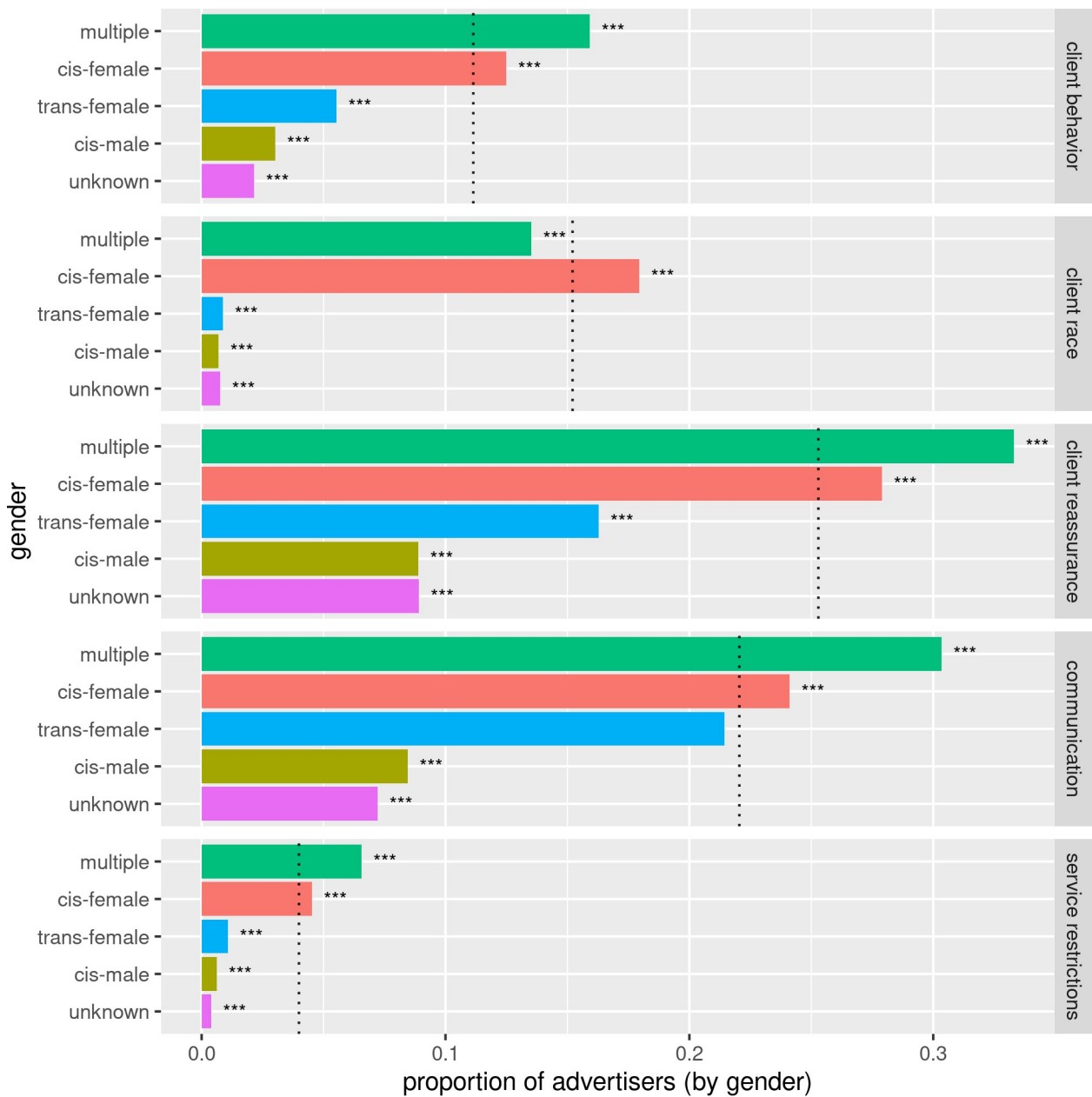

**Fig 6. Advertisers by gender associated with the top 5 codes.** The dotted lines indicate the proportion of all advertisers associated with that theme. Most genders were significantly different from the overall proportion for each theme: *** p < 0.001.

(*restriction*|Asian Canadian) = 0.09, 95% CI [-0.30–0.12], p = 0.00125; p(*restriction*|Indo Canadian) = 0.23, 95% CI [-0.12–0.03], p = 0.001738). The only theme that was less probable for Black advertisers in 2021–2022 was *client race* (p(*client race*|Black) = 0.11; p(*client race*|not-Black) = 0.14).

Moorman and Harrison, in a sample of 561 Detroit, Michigan Backpage.com ads collected in 2012–2013 reported similar patterns of risk management messaging. Their descriptions of messages matched themes of *communication* (two messages 24–25% ads), *client behavior* (three messages 23–70% ads), *pimps or law enforcement* (9% ads), *client age* (2.2% ads), and

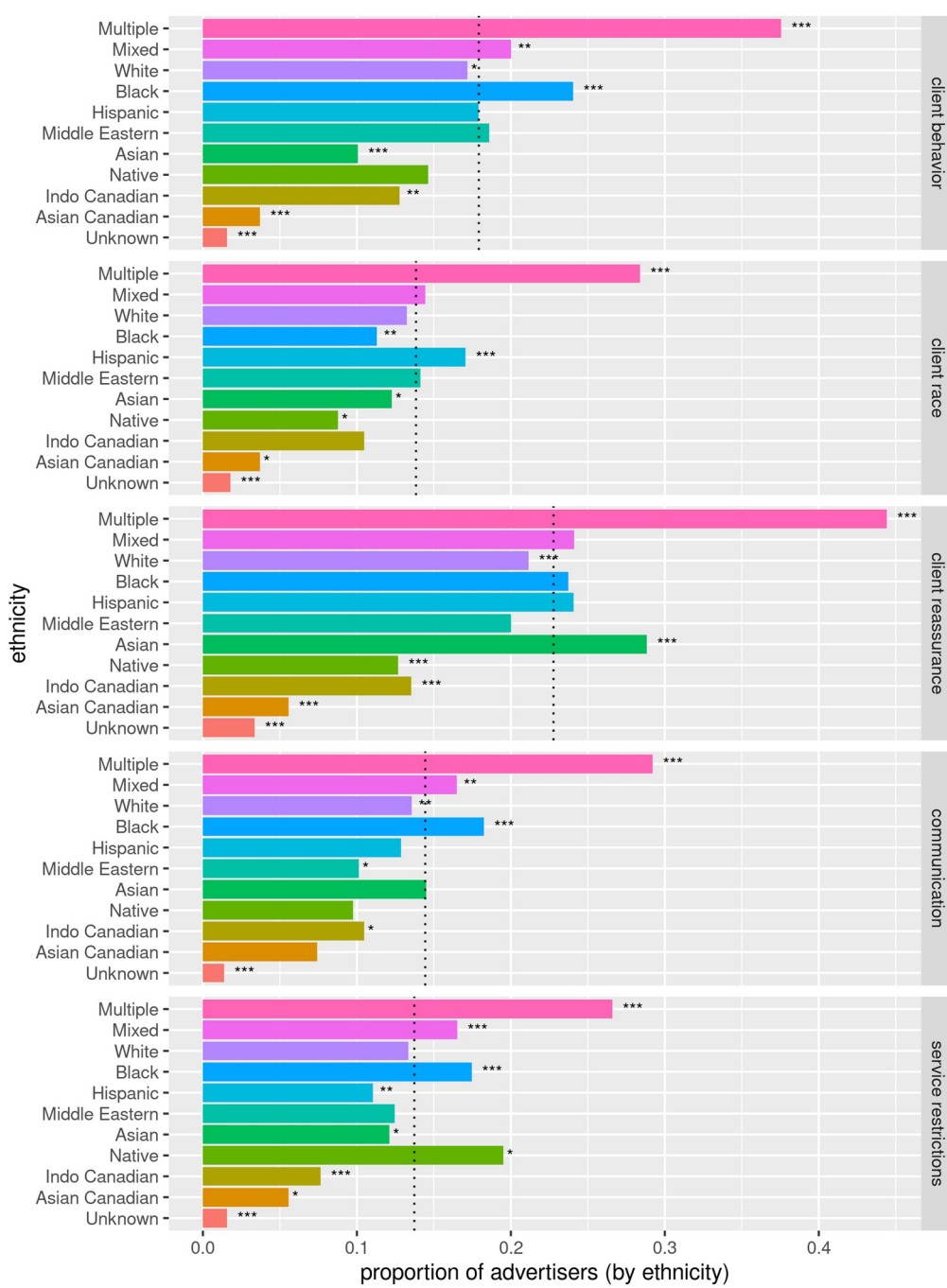

**Fig 7. 2021–2022 advertisers associated with the top 5 codes by ethnicity.** *Mixed* in this context is a single ethnic option, but *Multiple* refers to advertisers who indicated more than one ethnicity in their ads. The dotted lines indicate the proportion of all 2021–2022 advertisers associated with that theme. Most ethnicities were significantly different from the overall proportion for each theme: *** p < 0.001, ** p < 0.01, * p < 0.05.

*client race* (2.4% ads) [8] (see supporting information S4 File "moorman+harrison" for a comparison). Black advertisers used two of the *client behavior* messages ("no games" and "no fakers"), *client age*, and one of the *communication* messages ("no texting") more frequently than all advertisers. Canadian ad counts for these themes were found to range from the more

prevalent *communication* 17.9% (N = 757481), *client race* 7.2% (N = 305190), and *client behavior* 6.2% (N = 262554) themes, to the least prevalent *pimps or law enforcement* 0.4% (N = 16040) and *client age* 0.1% (N = 6117) themes.

## Client race

Many advertisers restricted clients based on racial background. In the later collections, 32610 advertisers were associated with the *client race* restriction (p(*client race*) = 0.15, N = 32610; p(*client race*|2014–2016) = 0.16, N = 27215; p(*client race*|2021–2022) = 0.12, N = 5397). In the 2007–2009 collection the number of advertisers using racial restrictions was minimal (N = 12). Bigrams associated with these restrictions overwhelmingly referred to Black or African American clients. Fig 8 summarizes the demographic characteristics of the advertisers associated with *client race*. Advertisers who were significantly more likely to be associated with the *client race* code were cis-female advertisers, advertisers who advertised in multiple provinces, and advertisers who only advertised in Ontario (p(*client race*|cis-female) = 0.18, N = 31540, 95% CI [0.03, 0.03], p < 0.001; p(*client race*|multiple provinces) = 0.25, N = 5992, 95% CI [0.10, 0.11], p < 0.001; p(*client race*|Ontario) = 0.23, N = 18353, 95% CI [0.08, 0.08], p < 0.001). Those exclusively identifying as cis-male were the least likely to be associated with *client race* restrictions (p(*client race*|cis-male) < 0.01, N = 101, *prop.test* CI [-0.15, -0.14], p < 0.001).

Ethnic self-identification affected the probability of being associated with the *client race* theme. Where data was available, the majority of advertisers (60%, N = 23429) did not identify as White and White advertisers were slightly but significantly less likely to be associated with *client race* restrictions (p(*client race*|non-White) = 0.14, N = 3333 versus p(*client race*|White) = 0.13, N = 2062, 95% CI [-0.017, -0.003], p = 0.007). *Multiple* ethnicity (0.28, N = 1811) and *Hispanic* (0.17, N = 309) advertisers were significantly more likely to be associated with *client race* than all advertisers. However, *Asian* (0.12, N = 309), *Black* (0.11, N = 209), *Native* (0.09, N = 18) and *Unknown* (0.02, N = 119) advertisers were all significantly less likely to be associated with this restriction.

A search for the terms "all races", "all nationalities", "all backgrounds", and "all ethnicities" found 2139 advertisers (1%). The number of advertisers using these terms increased by a factor of 7 from 2007–2009 (0.003, N = 27) to 2021–2022 (0.02, N = 874).

## Discussion and conclusions

Over the 15-year period represented by the collections in this study, roughly similar proportions of advertisers used the word "no". However, how advertisers used "no" in the earliest collection appears to be different from how "no" was used in later collections. In later collections, the word "no" was more likely to refer to restrictions: *client behavior*, *service restrictions*, or *client race*. Overall, around a quarter of all advertisers used a restriction at least once. Many advertisers who used restrictions did not use them consistently, suggesting that perception of risk may fluctuate on both small and large time scales. Cis-female advertisers were much more likely to use restrictions in ads. However, it was a surprise to find that this was not the case for trans female advertisers who historically have been subject to greater violence than other sex workers [55–57].

From 2014–2016 to 2021–2022 the top five themes remained the same with *client behavior* increasing from fourth to second rank and *communication*, *client race*, and *service restrictions* all converging on similar values (probability ∼ 0.14). However, data from British Columbia shows that, at least regionally, the relative prevalence of each theme could change significantly. In particular, the *client race* theme, which was not significant in 2007–2009 in BC, grew

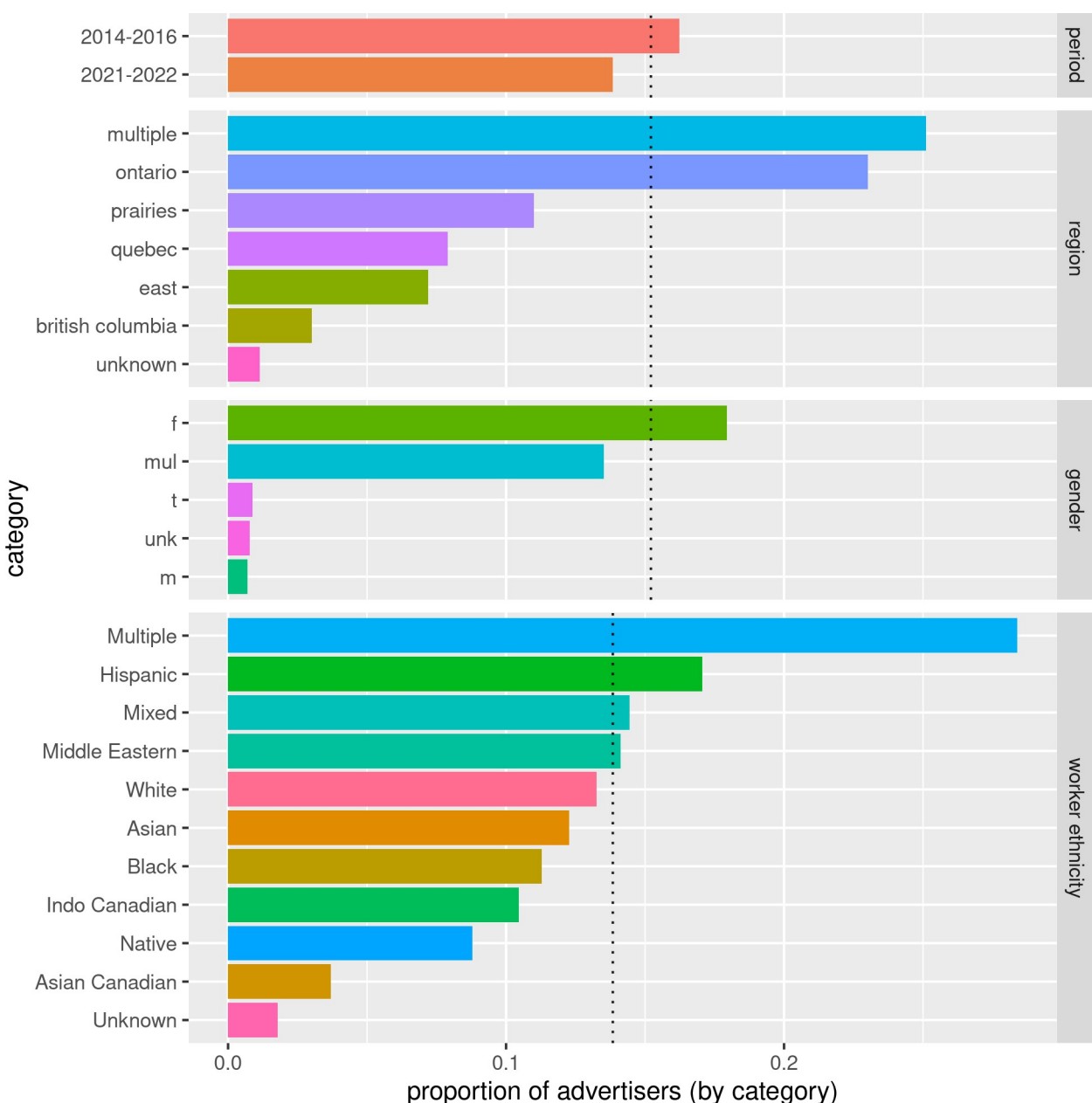

**Fig 8. Summary of advertisers by category who restrict clients based on *client race*.** The dotted lines are the proportion of all advertisers with this type of restriction. 2021–2022 was the only period with reliable data for ethnicity.

steadily in prevalence through 2014–2016 and 2021–2022. This was also the case for most themes with the exception of *communication*.

Condom refusal can be reduced when workers advertise online [20, 58, 59] and, while less common than other themes, the prevalence of the *service restrictions* theme increased over time. This increase might suggest that some advertisers became less tolerant of unhealthy sexual practices. A possible reason for this could be that nuisance behaviors by prospective clients have become more common for some advertisers. The small positive association between *client behavior* and *service restrictions* may be indicative of this.

The Moorman and Harrison study [8] showed similar thematic patterns to this study suggesting that the main themes are generally relevant. The evidence presented here shows that in Canada, while Black and Mixed ethnicity advertisers used significantly more restrictions, some ethnicities such as Asian, Asian Canadian, and Indo Canadian used significantly less. Statements regarding client age and law enforcement also appeared to be more frequent in Moorman and Harrison.

Sex workers should always have the right to refuse service [60, 61] and even in a criminalized environment, as is the case in Canada [36], sex workers routinely exercise agency [5, 62–65]. Nevertheless, it was striking that over 32000 advertisers used *client race* restrictions, a phenomenon also seen in Moorman and Harrison. This type of restriction appears to be mainly associated with cis-female advertisers, and appears to be more prominent in some regions. Notably, most advertisers using this restriction did not self-identify as White and 11% of advertisers who self-identified as Black were associated with it. The strong association between locale and gender in this case provides further evidence that the experiences of sex workers are highly dependent on identity and personal circumstances. Advertisers rarely described why they employed this restriction and further research is needed to understand the reasons for it.

## Limitations

While an archival study such as this cannot determine why advertisers used language the way they did, "no" statements relating to restrictions are assumed to reflect perceptions of risk and the ad collections, while likely not complete, are assumed to be representative. Typically about 5% of advertisers are not contact sex workers. While the variables identified in ads may not reflect all the relevant variables for understanding perceptions of risk they are assumed to be relevant. Advertisers can be workers or third parties and language use by third parties may not reflect the attitudes of the workers they represent.

The terminology described here may be specific to ads and there may be other ways in which advertisers indicate similar meanings to the themes discussed in this study. The estimated probabilities should be considered a lower bound for these types of messages. For example, the way similar types of statements are represented in French language ads can be different from how these are represented in English, resulting in undercounting of Francophone advertisers. This could be seen in the 2014–2016 data, where one predominantly Francophone site had a far smaller proportion of included advertisers (see Site 4 in the supporting information S3 File).

## Conclusions

The use of the word "no" in ads reveals a number of trends that warrant further exploration. In particular, the *client race* restriction seen in this study has not been reported widely. While Moorman and Harrison reported the use of the statement "No Black men", it was surprising that they and others citing them [20, 39, 40, 66–72] did not pursue the topic further. Although there has been recent work on the experiences of clients generally [73–75], there appears to be a lack of research on the specific experience of non-White clients.

Legally, client restrictions introduce a potential conflict between consent as defined in section 273.1 of the Canadian Criminal Code [76], applicable at any time including prior to meeting in person, and discrimination based on race and age protected by the Canadian Human Rights code in sections 5 and 12 [77]. In an environment where clients are criminalized it may be easy to ignore these conflicts. However, in a decriminalized context human rights policy may need to be updated to explicitly give workers the right to refuse service for any reason.

Online advertising that supports sex workers is an essential structural tool. Many workers who advertise online use restrictions to protect themselves. These restrictions are a reflection of sex worker agency and are an important source of information on how stigma manifests itself in the industry. The evidence presented here shows that the number of advertisers who use these restrictions is increasing. This growing group of Canadian sex workers have become less tolerant of bad behavior online and have become more assertive about their rights and sexual health, even in the face of significant structural challenges.

## Supporting information

**S1 Appendix. Craigslist usage in Vancouver 2007–2009.** https://osf.io/ys5ed.
(DOCX)

**S2 Appendix. Effect of weighting on probability estimates.** https://osf.io/wdj3t.
(DOCX)

**S1 File. MariaDB database.** https://osf.io/7s69q.
(ZIP)

**S2 File. QualCoder project with coded documents.** https://osf.io/k85pf.
(ZIP)

**S3 File. Spreadsheet showing contribution of individual sites.** https://osf.io/rspv4.
(XLSX)

**S4 File. Tables.** https://osf.io/2svka.
(XLSX)

**S1 Fig. Comparison of "no" advertisers by variable value.** https://osf.io/8k234.
(PNG)

## Acknowledgments

Sadie Hudson (http://www.answersociety.org/) and Robin C. A. Barrett provided editorial input on various versions of this manuscript.

## Author Contributions

**Conceptualization:** Lynn Kennedy.

**Data curation:** Lynn Kennedy.

**Formal analysis:** Lynn Kennedy.

**Funding acquisition:** Lynn Kennedy.

**Investigation:** Lynn Kennedy.

**Methodology:** Lynn Kennedy.

**Project administration:** Lynn Kennedy.

**Resources:** Lynn Kennedy.

**Software:** Lynn Kennedy.

**Supervision:** Lynn Kennedy.

**Validation:** Lynn Kennedy.

**Visualization:** Lynn Kennedy.

**Writing – original draft:** Lynn Kennedy.

**Writing – review & editing:** Lynn Kennedy.

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
