## [Decision Letter · Decision Letter 0]

20 Dec 2023

PONE-D-23-33995The Changing Meaning Of “No” In Canadian Sex WorkPLOS ONE

Dear Dr. Kennedy,

Thank you for submitting your manuscript to PLOS ONE. After careful consideration, we feel that it has merit but does not fully meet PLOS ONE’s publication criteria as it currently stands. Therefore, we invite you to submit a revised version of the manuscript that addresses the points raised during the review process.

We look forward to receiving your revised manuscript.

Kind regards,

Vanessa Carels

Staff Editor

PLOS ONE

Journal Requirements:

2. We are unable to open your Supporting Information file [S1 File]. Please kindly revise as necessary and re-upload.

Reviewers' comments:

Reviewer's Responses to Questions

**Comments to the Author**

1. Is the manuscript technically sound, and do the data support the conclusions?

Reviewer #1: Partly

Reviewer #2: Yes

Reviewer #3: Partly

2. Has the statistical analysis been performed appropriately and rigorously? 

Reviewer #1: Yes

Reviewer #2: Yes

Reviewer #3: Yes

3. Have the authors made all data underlying the findings in their manuscript fully available?

Reviewer #1: Yes

Reviewer #2: Yes

Reviewer #3: Yes

4. Is the manuscript presented in an intelligible fashion and written in standard English?

Reviewer #1: No

Reviewer #2: Yes

Reviewer #3: Yes

5. Review Comments to the Author

Reviewer #1: Dear author,

I appreciate the opportunity to review your interesting research. This analysis has the potential to make an important contribution to the literature on sex worker advertising. However, the manuscript has many limitations that limit its potential contribution. In general, the manuscript needs more contextual information about sex work criminalization in Canada and how this impacts sex workers' ads and communications, has flaws regarding differences between comparison groups which need to be justified and/or discussed as a limitation, and the discussion and conclusion sections feature some author interpretations of the findings that are unsubstantiated.

Introduction/Background

The current Background section features very minimal information about the context of sex work in Canada. Critically, there is no discussion of sex work criminalization, including the federal legislative shift (to the Protection of Communities and Exploited Persons Act which took place in 2014 and explicitly criminalized the purchase of sex for the first time in Canadian history), which was during the time frame of the data utilized for this study. Sex work criminalization very heavily shapes the ways that sex workers advertise and communicate with clients. This context is essential background for such an analysis. Further, PLOS One is a general journal that publishes for a non-specialized audience. The author should not assume that the reader has a background in sex work research - more is needed to introduce sex work in Canada. The Background should also justify why this research is needed.

"As a safety strategy, advertising online significantly reduces risk for contact sex workers[3,17,18]. Workers describe investing considerable effort in getting to know clients before any in-person contact is initiated [3,5,19] and this can include more formal screening processes in some cases [3,18–21]. Information sharing between workers is another important safety strategy

[22] and how workers share information has changed as the internet has become the dominant way that workers advertise [3,19,22]. Communication in advertising plays an important part in this process as a critical first step in the worker-client relationship."

The above is excellent contextual information and well-cited. How safety and online advertising is linked to sex work criminalization in Canada is missing - this is needed for the reader.

Methodological considerations

Paragraph 1 - "The majority of these statements appeared to refer to restrictions. A review of older and newer datasets, included in this study, showed that this type of phraseology has been in use by sex work advertisers for a long time."

Why is this the case? Why do sex workers express limits like these in ads? It would be helpful to contextualize this by drawing on qualitative research which has examined it. Sex workers have never been able to overtly, directly, clearly communicate about their services via their advertising due to sex work criminalization. This context is very important, yet missing from this manuscript.

Research questions and objectives

I appreciate that this section is here, but should be more succinctly stated. Ie, 'This study aimed to explore 1) the use of the word 'no' over time, and 2) whether the use of 'no' differs by the demographic characteristics of the advertisers, in online sex work advertisements in Canada from 2007-2021.'

Materials and methods

"The likelihood of there being an association between an advertiser using “no” and a major theme is calculated for several variables,

including: time period, region, advertiser gender, and advertiser ethnicity."

This phrasing is unclear and should be checked with a statistician. From my understanding, this analysis is looking at odds ratios, so the word 'likelihood' is not an accurate way to communicate this.

Extracting ad data

"April 1, 2007 to March 31, 2009 inclusive, November 1, 2014 to December 31, 2016 inclusive, and September 15, 2021 to September 22, 2022 inclusive."

Why two two year periods then a one year period? This should be clarified and justified, and also cited as a limitation. The author should also justify the decision to include the 2007-2009 collection since it's focused on BC only. I understand the interest in examining use of 'no' over time, but the different time periods are not all assessing the same population. This is a limitation.

Results

Table 1 - Why were 6 websites used in the middle time period, but only 1 website for the first and third time periods? This should be clarified, justified, and cited as a limitation. The author should consider splitting this analysis into two or more different analyses to make the comparison groups more similar by place (BC vs all of Canada) and by websites used. The use of comparison groups that differ by more than one characteristic is a flaw in this study and needs significant justification.

The results section is extremely long and meandering. The author should consider the research objective, consider which are the most important results, and present only those.

Discussion and conclusions

"Cis-female advertisers were also much more likely to use language relating to restrictions in ads, perhaps reflecting the greater sense of risk experienced by these workers."

The second part of this sentence is an overreach by the author and is presented with no citations supporting it. Literature suggests that trans women also experience great risk and violence in sex work.

"This suggests that, as online advertising has become more common, contact sex workers may be under less pressure to accept unhealthy sexual practices."

This claim is unsubstantiated and an interpretative overreach. This research did not explore the prevalence of online advertising and pressure to accept 'unhealthy sexual practices'. Other research has examined factors shaping sex workers' agency, ability to negotiate condom use, etc - this should be referenced here. The author has made an assumption linking service restrictions to factors related to advertising, and this is not substantiated by their findings nor by citing other publications.

Conclusions

"Furthermore, collectives and advertisers who advertised in multiple provinces appear to have a heightened sense of risk. More research is needed to understand why this is the case..."

This statement represents another conclusion drawn by the author which unsubstantiated and an interpretative overreach. This study did not examine 'heightened sense of risk' or cite literature about this. What the author has characterized as a 'heightened sense of risk' could conversely be characterized as greater agency, evidenced by sex workers' ability to express restrictions in their ads, which the author alludes to in the next paragraph. The discussion and conclusion should be limited to what was found in this study's results, and should be more grounded in other literature. It's not appropriate for the author to suggest findings or conclusions which were not present in the study's data.

Minor comments:

Introduction, paragraph 1 - I am a sex work researcher and have never come across the term 'contact sex worker' in literature or in direct discussions with sex workers. Has this term been used or defined elsewhere? If so, please cite.

"Peace of mind refers to “no” statements that were intended to reassure prospective clients." - 'Peace of mind' could refer to either for client or provider. 'Reassurance for client' would be more clear.

Reviewer #2: This analysis is novel and interesting but the narrative around the usefulness and what this actually may indicate is not drawn out enough to make the article useful or a contribution to the literature. There are some easy fixes here. The general point, and this is relevant for the abstract and the introduction, is that it is not at all clear why 'no' is at all interesting. If anything is speaks to consent debates and behaviours. There needs to be more robust introduction and clarity around research questions and why this analysis is useful - and its longitudinal merits. Maybe more of a framing around what can adverts for sexual services tell us about changing practices in the the sex industry, or how sex workers control (attempt to) their work/conditions/clients.

The first few pages just had me thinking - what is the point of this research' - it was not clear. The title also is confusing.

Pge 4 - what about the limitations of using ad data from platforms - there needs to be a more critical discussion of this.

page 5 - line 80 - this is an important line but is incomplete - doesnt make sense

page 84 - phraseology - again you have not said why this is important

in methods - add more about the importance of longitudinal

the analysis detail is very comprehensive and robust

my other main concern is about the under-developed nature of the discussion.

is there much more to be made about how this data can really help us think about changes over time in sex work practices in Canada. It seems odd and remiss that the major changes and volatility in Canadian law over the past decade has not been a key part of the discussion. Obviously correlation cannot be proven but this legal landscape has a major influence on online practices (think sesta/fosta).

what are the implications more broadly outside of Canada ?

can more be said about how this data could suggest sex workers are controlling their work more / being more automous in their practices / using comms and marketing to keep themselves safe in a criminalised setting?

Reviewer #3: I think this was an interesting idea but I have a few queries.

1.) What is your theoretical framework?

2.) Why didn't you use qualitative methods- this might have provided a much needed context and detail, I feel as it is you rely on a lot of assumptions

3.) Why did you search 'no' and not other related terms, e.g. 'forbidden', 'not be tolerated', 'blocked', 'blacklisted?

4.) How can you be sure adverts are written by the individual? It could be a manager/group with a template profile so it seems bold to correlate between gender/ethnicity and the use of 'no'.

5.) What about factors beyond agency, e.g. could there bean increase in clients' boundary pushing/haggling/abusive behaviours? These could be explained by other factors, e.g. financial crisis, increased consumption of violent pornography. E.g. in the UK c. 30 years ago unprotected oral sex was extremely rare yet now is almost standard so those who do not offer this are sure to stipulate this on their profiles.

6.) Did you set out to explore multiples contexts of 'no', e.g. as reassurance for clients (e.g. 'no clock watching') If not, doesn't the use of non-restrictive 'no' dilute the agency argument?

7.) What is the significance of the time periods you choose?

8.) p.26-26 you state that client race restrictions, e.g. black clients are puzzling as black clients are not more sexist. Why is sexism the only factor you consider here? This is an unjustified assumption and doesn't interrogate sex workers' reasons for race restrictions, e.g. wanting to avoid the likelihood of meeting members of their own community, having had a traumatic experience in the past.

9.) It is a big assumption that 'no' relates to Me Too

10.) What are the implications of your findings, presuming the statement about increasing agency can be proven to be true?

6. PLOS authors have the option to publish the peer review history of their article (what does this mean?). If published, this will include your full peer review and any attached files.

Reviewer #1: No

Reviewer #2: No

Reviewer #3: No

---

## [Author Response · Author response to Decision Letter 0]

1 Feb 2024

Dear Dr. Kennedy,

Thank you for submitting your manuscript to PLOS ONE. After careful consideration, we feel that it has merit but does not fully meet PLOS ONE’s publication criteria as it currently stands. Therefore, we invite you to submit a revised version of the manuscript that addresses the points raised during the review process.

We look forward to receiving your revised manuscript.

Kind regards,

Vanessa Carels

Staff Editor

PLOS ONE

Journal Requirements:

2. We are unable to open your Supporting Information file [S1 File]. Please kindly revise as necessary and re-upload. - I have reformatted the file.

Reviewers' comments:

Reviewer's Responses to Questions

Comments to the Author

1. Is the manuscript technically sound, and do the data support the conclusions?

Reviewer #1: Partly

Reviewer #2: Yes

Reviewer #3: Partly

2. Has the statistical analysis been performed appropriately and rigorously?

Reviewer #1: Yes

Reviewer #2: Yes

Reviewer #3: Yes

3. Have the authors made all data underlying the findings in their manuscript fully available?

Reviewer #1: Yes

Reviewer #2: Yes

Reviewer #3: Yes

4. Is the manuscript presented in an intelligible fashion and written in standard English?

PLOS ONE does not copy edit accepted manuscripts, so the language in submitted articles must be clear, correct, and unambiguous. Any typographical or grammatical errors should be corrected at revision, so please note any specific errors here.

Reviewer #1: No

Reviewer #2: Yes

Reviewer #3: Yes

5. Review Comments to the Author

Reviewer #1: 

Dear author,

I appreciate the opportunity to review your interesting research. This analysis has the potential to make an important contribution to the literature on sex worker advertising. However, the manuscript has many limitations that limit its potential contribution. In general, the manuscript needs more contextual information about sex work criminalization in Canada and how this impacts sex workers' ads and communications, has flaws regarding differences between comparison groups which need to be justified and/or discussed as a limitation, and the discussion and conclusion sections feature some author interpretations of the findings that are unsubstantiated. - Thank you for taking the time to thoroughly review this work. I have added more contextual information. We should be careful to note that the study does not compare groups but instead looks at different dimensions such as gender, locale, etc. with respect to the use of language in ads. 

Introduction/Background

The current Background section features very minimal information about the context of sex work in Canada. Critically, there is no discussion of sex work criminalization, including the federal legislative shift (to the Protection of Communities and Exploited Persons Act which took place in 2014 and explicitly criminalized the purchase of sex for the first time in Canadian history), which was during the time frame of the data utilized for this study. Sex work criminalization very heavily shapes the ways that sex workers advertise and communicate with clients. This context is essential background for such an analysis. Further, PLOS One is a general journal that publishes for a non-specialized audience. The author should not assume that the reader has a background in sex work research - more is needed to introduce sex work in Canada. The Background should also justify why this research is needed. - I have added historical context to the Introduction.

"As a safety strategy, advertising online significantly reduces risk for contact sex workers[3,17,18]. Workers describe investing considerable effort in getting to know clients before any in-person contact is initiated [3,5,19] and this can include more formal screening processes in some cases [3,18–21]. Information sharing between workers is another important safety strategy

[22] and how workers share information has changed as the internet has become the dominant way that workers advertise [3,19,22]. Communication in advertising plays an important part in this process as a critical first step in the worker-client relationship."

The above is excellent contextual information and well-cited. How safety and online advertising is linked to sex work criminalization in Canada is missing - this is needed for the reader.

Methodological considerations

Paragraph 1 - "The majority of these statements appeared to refer to restrictions. A review of older and newer datasets, included in this study, showed that this type of phraseology has been in use by sex work advertisers for a long time."

Why is this the case? Why do sex workers express limits like these in ads? It would be helpful to contextualize this by drawing on qualitative research which has examined it. Sex workers have never been able to overtly, directly, clearly communicate about their services via their advertising due to sex work criminalization. This context is very important, yet missing from this manuscript. - This is a good question. I would disagree that sex workers have never been able to clearly, directly communicate about their services. They haven’t legally been able to do so in most countries but in practice in Canada many of the classified ads are quite graphic and Twitter even more so.

Research questions and objectives

I appreciate that this section is here, but should be more succinctly stated. Ie, 'This study aimed to explore 1) the use of the word 'no' over time, and 2) whether the use of 'no' differs by the demographic characteristics of the advertisers, in online sex work advertisements in Canada from 2007-2021.' - I have replaced the text using the wording you suggest.

Materials and methods

"The likelihood of there being an association between an advertiser using “no” and a major theme is calculated for several variables,

including: time period, region, advertiser gender, and advertiser ethnicity."

This phrasing is unclear and should be checked with a statistician. From my understanding, this analysis is looking at odds ratios, so the word 'likelihood' is not an accurate way to communicate this. - The measure used is not odds (ie p(x)/p(~x)) but probability. I have updated the document to make this clearer.

Extracting ad data

"April 1, 2007 to March 31, 2009 inclusive, November 1, 2014 to December 31, 2016 inclusive, and September 15, 2021 to September 22, 2022 inclusive."

Why two two year periods then a one year period? This should be clarified and justified, and also cited as a limitation. The author should also justify the decision to include the 2007-2009 collection since it's focused on BC only. I understand the interest in examining use of 'no' over time, but the different time periods are not all assessing the same population. This is a limitation. - Data availability was the reason. I have added an explanation of why the data was included. The decision to include the 2007-2009 data is partly because there is very little information on this period. I have added an appendix that looks at how the number of workers grew during this period. This is linked to the historical discussion in the introduction.

Results

Table 1 - Why were 6 websites used in the middle time period, but only 1 website for the first and third time periods? This should be clarified, justified, and cited as a limitation. The author should consider splitting this analysis into two or more different analyses to make the comparison groups more similar by place (BC vs all of Canada) and by websites used. The use of comparison groups that differ by more than one characteristic is a flaw in this study and needs significant justification. - I have added text explaining why I have not broken out analyses by site and region in the Methods section and explained why there were 6 sites at one period in the historical context in the Introduction. The fact that all of the “no” statement themes were still prevalent in 2022 suggests that there are certain invariants over time. I have added a spreadsheet showing the contribution of each site (which roughly corresponds to time period) including a breakdown by gender for the top five codes for interested readers. Note that all the data used to generate the reported findings is available and readers are encouraged to look at the source data also available on request.

The results section is extremely long and meandering. The author should consider the research objective, consider which are the most important results, and present only those. - Even in an overview such as this there is a lot to cover. I have removed most of the tables and edited the text. I hope this makes this section easier to read. However, I added a section comparing the results to those of an earlier study (Moorman & Harrison, 2016) as this is relevant to the findings.

Discussion and conclusions

"Cis-female advertisers were also much more likely to use language relating to restrictions in ads, perhaps reflecting the greater sense of risk experienced by these workers."

The second part of this sentence is an overreach by the author and is presented with no citations supporting it. Literature suggests that trans women also experience great risk and violence in sex work. - I have removed the last part of the sentence. The differences in how trans women and cis women communicate was surprising.

"This suggests that, as online advertising has become more common, contact sex workers may be under less pressure to accept unhealthy sexual practices."

This claim is unsubstantiated and an interpretative overreach. This research did not explore the prevalence of online advertising and pressure to accept 'unhealthy sexual practices'. Other research has examined factors shaping sex workers' agency, ability to negotiate condom use, etc - this should be referenced here. The author has made an assumption linking service restrictions to factors related to advertising, and this is not substantiated by their findings nor by citing other publications. - This is a good point and I have updated the discussion to reflect this. I looked at how correlated the main themes were and there is a small correlation between client behavior and service restrictions which might support increasing pressure from clients.

Conclusions

"Furthermore, collectives and advertisers who advertised in multiple provinces appear to have a heightened sense of risk. More research is needed to understand why this is the case..."

This statement represents another conclusion drawn by the author which is unsubstantiated and an interpretative overreach. This study did not examine 'heightened sense of risk' or cite literature about this. What the author has characterized as a 'heightened sense of risk' could conversely be characterized as greater agency, evidenced by sex workers' ability to express restrictions in their ads, which the author alludes to in the next paragraph. The discussion and conclusion should be limited to what was found in this study's results, and should be more grounded in other literature. It's not appropriate for the author to suggest findings or conclusions which were not present in the study's data. - I have removed this sentence.

Minor comments:

Introduction, paragraph 1 - I am a sex work researcher and have never come across the term 'contact sex worker' in literature or in direct discussions with sex workers. Has this term been used or defined elsewhere? If so, please cite. - The distinction was suggested by the sex worker who reviewed the paper. I’ve made the concept clearer in the introduction. The distinction is important as some forms of sex work are subject to more legal sanctions in Canada than others.

"Peace of mind refers to “no” statements that were intended to reassure prospective clients." - 'Peace of mind' could refer to either for client or provider. 'Reassurance for client' would be more clear. - “Peace of mind” was changed to “client reassurance” which I hope will clarify the concept.

Reviewer #2: 

This analysis is novel and interesting but the narrative around the usefulness and what this actually may indicate is not drawn out enough to make the article useful or a contribution to the literature. There are some easy fixes here. The general point, and this is relevant for the abstract and the introduction, is that it is not at all clear why 'no' is at all interesting. If anything it speaks to consent debates and behaviours. There needs to be more robust introduction and clarity around research questions and why this analysis is useful - and its longitudinal merits. Maybe more of a framing around what adverts for sexual services tell us about changing practices in the sex industry, or how sex workers control (attempt to) their work/conditions/clients. - I have added language on why this research makes a contribution to the literature to the abstract and introduction: there is little research that reviews a large number of documents representing large numbers of advertisers. Also, this study found similar patterns of language use as a previous study (Moorman & Harrison, 2016).

The first few pages just had me thinking - what is the point of this research' - it was not clear. The title also is confusing. - The point is to better understand how sex workers communicate and show that on a larger scale patterns seen in previous research can apply. In particular, how common is risk mitigating language and who uses it? The previous studies that I am aware of have generally used limited document samples. This study takes a different approach to see what else can be understood from these archival sources. Maybe a more precise wording of the title would be “Understanding the usage of the word “no” in Canadian sex work advertising”. The wording used is easier to read however. 

Pge 4 - what about the limitations of using ad data from platforms - there needs to be a more critical discussion of this. - I discuss this in the Introduction, Methods and Limitations section of the Discussion. 

page 5 - line 80 - this is an important line but is incomplete - doesnt make sense - I have clarified which study I am referring to here.

page 84 - phraseology - again you have not said why this is important

in methods - add more about the importance of longitudinal - Do you mean line 84? I have added text to describe the rationale for treating time-period as a variable. It is important to see what is invariant as well as what has changed.

the analysis detail is very comprehensive and robust - thank you

my other main concern is about the under-developed nature of the discussion.

is there much more to be made about how this data can really help us think about changes over time in sex work practices in Canada. It seems odd and remiss that the major changes and volatility in Canadian law over the past decade has not been a key part of the discussion. Obviously correlation cannot be proven but this legal landscape has a major influence on online practices (think sesta/fosta). - I have added to the introduction background on the legal changes that coincided with the rise of online advertising.

what are the implications more broadly outside of Canada ? - This is a good question and I hope there will be more research on the topic of race both from the perspective of workers and clients. The (Moorman & Harrison, 2016) study from Detroit Michigan found similar themes. The prevalence of risk related messaging is similar and Black and Mixed advertisers were more likely to use it. Unexpectedly, trans-female, Asian, and Indo-Canadian advertisers were less likely to use risk related messaging. 

can more be said about how this data could suggest sex workers are controlling their work more / being more automous in their practices / using comms and marketing to keep themselves safe in a criminalised setting? - It is beyond the scope of this paper to say whether advertisers are becoming more autonomous. I would think barriers to advertising (technical and financial) are eroding this. I discuss this in more detail in (Kennedy, 2023b). 

Reviewer #3: 

I think this was an interesting idea but I have a few queries.

1.) What is your theoretical framework? - I have added a paragraph discussing this to the Introduction before “Research questions and objectives”. 

2.) Why didn't you use qualitative methods- this might have provided a much needed context and detail, I feel as it is you rely on a lot of assumptions - I did use qualitative methods but in a different way from other research which allows for the inclusion of a much larger sample of documents. As with any archival research, follow up studies are needed to clarify why advertisers communicate in the way they do. The motivation was to do a study on this topic that did not simply repeat the methods of other studies. As with any research the choices made will limit the findings. 

3.) Why did you search 'no' and not other related terms, e.g. 'forbidden', 'not be tolerated', 'blocked', 'blacklisted? - I do discuss other safety related terms. However, the word “no” is used because it is very common, meaningful, and tends to be used in consistent ways. For example, the terms “forbidden”,”not be tolerated” and “blacklisted” were relatively uncommon, showing up in < 1% of the documents in each collection (note that “Blocked” also occurs in “no blocked …” so it is more common). Limiting the analysis to phrases including “no” makes a review of over 4 million documents feasible. However, there are limitations to this approach which I touch on in the discussion. In particular this analysis will tend to exclude ads exclusively in French although many of these include English text.

4.) How can you be sure adverts are written by the individual? It could be a manager/group with a template profile so it seems bold to correlate between gender/ethnicity and the use of 'no'. - I do not assume advertisers represent individuals. Previous work shows that roughly 50% of advertisers could be considered independents (Kennedy, 2022, 2023a, 2023b). I also treat advertisers who do not advertise using consistent gender, ethnicity etc. separately for this reason. 

5.) What about factors beyond agency, e.g. could there be an increase in clients' boundary pushing/haggling/abusive behaviours? These could be explained by other factors, e.g. financial crisis, increased consumption of violent pornography. E.g. in the UK c. 30 years ago unprotected oral sex was extremely rare yet now is almost standard so those who do not offer this are sure to stipulate this on their profiles. - This is beyond the scope of this paper naturally. I found that advertisers sometimes did discuss motivations for restricting clients, for example, but this was very rare. The very few who do speak of having bad experiences. However, this is a very interesting question and I hope researchers will begin to address the experience of clients more which might help provide clarity. There may also be significant cultural differences between workers that, with migration, might factor into this phenomenon.

6.) Did you set out to explore multiple contexts of 'no', e.g. as reassurance for clients (e.g. 'no clock watching') If not, doesn't the use of non-restrictive 'no' dilute the agency argument? - I set out to understand how restrictions are worded in the context of seeing many advertisers using “no Black gents” in ads and client reassurance emerged as an important theme. The fact that the prevalence of restrictions increased sharply from 2014-2016 to 2021-2022 I would say is evidence for agency as advertisers potentially lose money when they use them in ads. However, we should also be careful to note that not all advertisers use this language. Looking at how correlated the themes are I think the case could be made that there are distinct subgroups of advertisers, some of whom are more strict than others. Therefore making a general statement about all advertisers might be premature. I have updated the Conclusions to reflect this.

7.) What is the significance of the time periods you choose? - Historically 2007-2009 was dominated by “Site 1” (craigslist), 2014-2016 was dominated by “Site 2” (backpage) when craigslist had to close the erotic services section, and 2021-2022 was dominated by “Site 3” when backpage was seized by the FBI. I don’t mention who Site 3 is as they are still active. I have added a spreadsheet to the supplemental materials that breaks out the contribution of each website.

8.) p.26-26 you state that client race restrictions, e.g. black clients are puzzling as black clients are not more sexist. Why is sexism the only factor you consider here? This is an unjustified assumption and doesn't interrogate sex workers' reasons for race restrictions, e.g. wanting to avoid the likelihood of meeting members of their own community, having had a traumatic experience in the past. - I have removed this section. One of my collaborators mentioned that 20-30 years ago it was quite common if the worker had a Black partner that they wouldn’t see Black clients. It is hard to say if that is the case now. Generally, there seems to be little information on the experience of Black clients.

9.) It is a big assumption that 'no' relates to Me Too - I have removed the reference but I do believe that how the advertisers think would likely influence/be influenced by #MeToo. I would say the data shows they lead rather than follow the trend.

10.) What are the implications of your findings, presuming the statement about increasing agency can be proven to be true? - I have removed the statement about increasing agency. However, I do discuss the legal implications of excluding clients based on race, gender identity, and age in the context of Canadian law. There is a potential conflict between consent and discrimination in the law as it is currently worded. 

References:

Kennedy, L. (2022). The silent majority: The typical Canadian sex worker may not be who we think. PloS One, 17(11), e0277550–e0277550.

Kennedy, L. (2023a). Estimating turnover and industry longevity of Canadian sex workers. SocArXiv. https://doi.org/10.31235/osf.io/qr75c

Kennedy, L. (2023b). What was the effect of PCEPA on Canadian sex work advertising? SocArXiv. osf.io/preprints/socarxiv/87u29

Moorman, J. D., & Harrison, K. (2016). Gender, Race, and Risk: Intersectional Risk Management in the Sale of Sex Online. The Journal of Sex Research, 53(7), 816–824. https://doi.org/10.1080/00224499.2015.1065950

---

## [Decision Letter · Decision Letter 1]

19 Mar 2024

The Changing Meaning Of “No” In Canadian Sex Work

PONE-D-23-33995R1

Dear Dr. Kennedy,

We’re pleased to inform you that your manuscript has been judged scientifically suitable for publication and will be formally accepted for publication once it meets all outstanding technical requirements.

Kind regards,

Emily Lund

Academic Editor

PLOS ONE

Additional Editor Comments (optional):

I have reviewed the revisions and, in agreement with Reviewer 2 and in contrast with Reviewer1, I believe that the authors have adequately addressed the previous reviewer comments.

Reviewers' comments:

Reviewer's Responses to Questions

**Comments to the Author**

1. If the authors have adequately addressed your comments raised in a previous round of review and you feel that this manuscript is now acceptable for publication, you may indicate that here to bypass the “Comments to the Author” section, enter your conflict of interest statement in the “Confidential to Editor” section, and submit your "Accept" recommendation.

Reviewer #1: (No Response)

Reviewer #2: All comments have been addressed

2. Is the manuscript technically sound, and do the data support the conclusions?

Reviewer #1: Yes

Reviewer #2: Yes

3. Has the statistical analysis been performed appropriately and rigorously? 

Reviewer #1: Yes

Reviewer #2: Yes

4. Have the authors made all data underlying the findings in their manuscript fully available?

Reviewer #1: Yes

Reviewer #2: Yes

5. Is the manuscript presented in an intelligible fashion and written in standard English?

Reviewer #1: Yes

Reviewer #2: Yes

6. Review Comments to the Author

Reviewer #1: Dear author,

I appreciate the opportunity to re-review the manuscript. My most significant feedback in my initial thorough review was that this manuscript was lacking in contextual information about the legislative setting in Canada with regard to sex work criminalization and how this impacts communications and advertising. This feedback was also echoed by other reviewers, and your response mentions adding this background information, but I don't see it and don't feel that this feedback was adequately taken into account. Instead of introducing the reader to your subject, the Canadian setting, and the need for your study, the introduction now immediately describes 'Research at Scale' and 'Limits and Safety', the importance of which are unclear because the subject matter of the article is not adequately introduced. There is virtually no information about the context of sex work in Canada. PLOS One publishes for a general, non-specialized audience, and this manuscript requires far more background information for an audience not familiar with sex work. I appreciate the clarification regarding the Methods, but the Results section continues to be extremely long and meandering. This manuscript would be much stronger with a more robust background section and a more limited objective and results section. Another reviewer mentioned that linking MeToo to this research was an overreach, you stated that you had removed the reference, but the reference remains in the manuscript. I don't believe that the updated manuscript adequately reflects the thoughtful and extensive feedback that reviewers shared, and as a result, I don't believe it's ready for publication.

Reviewer #2: This article is much improved with a close reading of the reviewers comments and a refocus in how the data and conclusions are presented.

7. PLOS authors have the option to publish the peer review history of their article (what does this mean?). If published, this will include your full peer review and any attached files.

Reviewer #1: No

Reviewer #2: No

---

## [Editor Report · Acceptance letter]

26 Mar 2024

PONE-D-23-33995R1 

PLOS ONE

Dear Dr. Kennedy, 

I'm pleased to inform you that your manuscript has been deemed suitable for publication in PLOS ONE. Congratulations! Your manuscript is now being handed over to our production team.

Kind regards, 

on behalf of

Dr. Emily Lund 

Academic Editor

PLOS ONE